# Short- and Long-Term Effectiveness of Low-Level Laser Therapy Combined with Strength Training in Knee Osteoarthritis: A Randomized Placebo-Controlled Trial

**DOI:** 10.3390/jcm11123446

**Published:** 2022-06-15

**Authors:** Martin Bjørn Stausholm, Ingvill Fjell Naterstad, Patricia Pereira Alfredo, Christian Couppé, Kjartan Vibe Fersum, Ernesto Cesar Pinto Leal-Junior, Rodrigo Álvaro Brandão Lopes-Martins, Jon Joensen, Jan Magnus Bjordal

**Affiliations:** 1Department of Global Public Health and Primary Care, University of Bergen, 5009 Bergen, Norway; naterstad@gmail.com (I.F.N.); kjartan.fersum@uib.no (K.V.F.); ernesto.junior@uib.no (E.C.P.L.-J.); jon.joensen@uib.no (J.J.); jan.bjordal@uib.no (J.M.B.); 2Department of Physiotherapy, Occupational Therapy and Speech Therapy, School of Medicine, University of Sao Paulo, São Paulo 05508-070, Brazil; patriciaalfredo@yahoo.com.br; 3Physical and Occupational Therapy Research Unit, Bispebjerg and Frederiksberg University Hospital, 2400 Copenhagen, Denmark; christian.couppe@regionh.dk; 4Laboratory of Phototherapy and Innovative Technologies in Health, Post-Graduate Program in Rehabilitation Sciences, Nove de Julho University, São Paulo 01504-001, Brazil; 5Research Group of Biophotonics and Experimental Therapeutics in Health and Esthetics-Post Graduate Program in Human Movement and Rehabilitation, University Center of Anápolis, Anápolis 75083-515, Brazil; ralopesmartins@gmail.com

**Keywords:** inflammation, knee osteoarthritis, low-level laser therapy (LLLT), strength training

## Abstract

Background: Both physical activity and low-level laser therapy (LLLT) can reduce knee osteoarthritis (KOA) inflammation. We conducted a randomized clinical trial to investigate the short- and long-term effectiveness of LLLT combined with strength training in persons with KOA. Methods: Fifty participants were randomly divided in two groups, one with LLLT plus strength training (*n* = 26) and one with placebo LLLT plus strength training (*n* = 24). LLLT and strength training were performed triweekly for 3 and 8 weeks, respectively. In the laser group, 3 joules 904 nm wavelength laser was applied to fifteen points (45 joules) per knee per session. Patient-reported outcomes, physical tests, and ultrasonography assessments were performed at baseline and 3, 8, 26, and 52 weeks after initial LLLT or placebo therapy. The primary outcomes were pain on movement, at rest, at night (Visual Analogue Scale), and globally (Knee injury and Osteoarthritis Outcome Score (KOOS) subscale). Parametric data were assessed with analysis of variance using Šidák’s correction. Results: There were no significant between-group differences in the primary outcomes. However, in the laser group there was a significantly reduced number of participants using analgesic and non-steroidal anti-inflammatory drugs and increased performance in the sit-to-stand test versus placebo-control at week 52. The joint line pain pressure threshold (PPT) improved more in the placebo group than in the laser group, but only significantly at week 8. No other significant treatment effects were present. However, pain on movement and joint line PPT were worse in the placebo group at baseline, and therefore, it had more room for improvement. The short-term percentage of improvement in the placebo group was much higher than in similar trials. Conclusions: Pain was reduced substantially in both groups. LLLT seemed to provide a positive add-on effect in the follow-up period in terms of reduced pain medication usage and increased performance in the sit-to-stand test.

## 1. Introduction

Knee osteoarthritis (KOA) is a progressive, disabling degenerative disease highly prevalent in the elderly population [1]. The disorder is driven by interactions between tissue damage, dysfunctional metabolism, and inflammation, and is associated with muscle weakness, pain, disability, and reduced quality of life (QoL) [1]. Inflammatory cells and humoral inflammatory mediators can trigger a release of matrix metalloproteinases by chondrocytes, leading to accelerated cartilage destruction [2]. In KOA, a higher level of inflammation is associated with more severe pain and rapid structural disease progression [1,2]. Therefore, the use of anti-inflammatory treatments in osteoarthritis is advised [1,2].

Low-level laser therapy (LLLT) is a safe intervention that has been found to reduce osteoarthritis inflammation in animal studies [3,4,5,6]. Furthermore, in vivo results of a meta-analysis by Xiang et al. (2017) demonstrated that laser therapy may have a protective effect on osteoarthritis cartilage, but only when applied with relatively low intensity (<1000 mW/cm^2^) [7]. Nevertheless, LLLT is not unconditionally recommended in dominating osteoarthritis treatment guidelines [8,9]. In the latest guideline by the Osteoarthritis Research Society International, LLLT is recommended at level 3 for KOA, but only in patients with a cardiovascular disorder, a gastrointestinal disorder, and/or a history of adverse events when using non-steroidal anti-inflammatory drugs (NSAIDs) [10]. We published a systematic review with a meta-analysis of randomized placebo-controlled trials in late 2019 and concluded that LLLT is capable of reducing short-term KOA pain [11]. The reviewed trials were subgrouped in terms of LLLT dose per treatment spot in adherence and nonadherence to the World Association for Laser Therapy treatment recommendations [12,13]. The recommended doses provided a substantial pain relief beyond placebo at therapy weeks 4–8 and at follow-ups 2–8 weeks after completed therapy. The non-recommended (lower) doses provided no or little positive effect [11]. However, there is a lack of evidence regarding the long-term effectiveness of LLLT in KOA; it was only investigated in three of the included trials [11,14,15,16]. Results of prior LLLT KOA systematic reviews are conflicting, but they featured no valid dose-response meta-analysis [17,18].

It is widely recommended that persons with KOA perform physical exercises [8,9], since they can reduce knee inflammation, although to a lesser extent than NSAIDs and LLLT [3,4]. A systematic review of high methodological quality by Bartholdy et al. (2017) indicates that exercise interventions following the American College of Sports Medicine (ACSM) definition of strength training is superior in increasing leg extension strength compared with different physical exercise regimens in KOA [19]. The ACSM recommends performing a minimum of two strength-training sessions weekly, comprising 2–4 sets with 8–12 repetitions maximum (RM) [20]. We conducted a systematic search for reports of trials on the current topic [11] and found that the effectiveness of LLLT as a supplement to an ACSM strength-training program in KOA had only been investigated in a few placebo-controlled randomized clinical trials (RCTs), and that these trials lacked long-term evaluations [21,22]. Furthermore, inflammatory markers were only assessed in two of the RCTs included in the review, and they only involved short-term evaluations [21,23]. Therefore, we decided to investigate both the short- and long-term effectiveness of LLLT associated with an ACSM strength training regimen in persons with painful KOA in a placebo-controlled RCT. The primary outcome was pain, since this is the dominating symptom in KOA [24].

## 2. Materials and Methods

### 2.1. Methods and Design

The RCT protocol was ethically approved by the Research Ethical Committee North Tromsø (reference: 2017/2417), registered on the website ClinicalTrials.gov (reference: NCT03750279), and published in a peer-reviewed journal (MDPI *Methods and Protocols*) [25]. The RCT was reported in line with the Consolidated Standards of Reporting Trials guidelines.

### 2.2. Participants

The subjects for the trial were recruited from the municipality of Bergen in Norway via written and verbal advertisement to the university outpatient clinic.

The inclusion criteria were persons of any gender, ≥50 years of age, unilateral or bilateral knee pain during movement corresponding to an intensity of ≥40 mm measured on the Visual Analogue Scale (VAS), knee pain in the last 3 months, and a KOA diagnosis established using the American College of Rheumatology clinical criteria [26]. The exclusion criteria were total meniscectomy, knee arthroplasty, corticosteroid treatment within the last 6 months, rheumatoid arthritis, cancer, severe cognitive impairment, neurological deficit in the leg, inability to speak and understand both English and Nordic languages, and lack of signed informed consent.

## 3. Procedure

### 3.1. Randomization

The participants in the trial were randomly allocated to one of two parallel groups with an allocation ratio of 1:1. One of the groups performed strength training and received LLLT and the other group performed strength training and received placebo LLLT. The randomization was performed after the baseline assessment by drawing concealed opaque envelopes containing a red or green label (group code) to conceal the allocation. The envelopes were prepared by a receptionist who was not otherwise involved in the study.

### 3.2. Strength Training

The participants performed exercises triweekly for the first 8 weeks. The exercises were performed under supervision of a physiotherapist in the university outpatient clinic three times per week in the first 3 weeks and once per week in the subsequent 5 weeks of the study (14 supervised and 10 unsupervised exercise sessions). The exercise program was designed by our research group. The program did not involve special equipment, except for an elastic band, which was distributed to the participants. This made the exercise program feasible to perform at home. Each session consisted of 5 min of warm up with light weight-bearing exercises for the lower limbs, followed by strength training on level 1 or 2. The participants completed strength training on level 1 in the first session and were allowed to interchange between the two levels in the subsequent sessions, that is, if this was recommended by the physiotherapist who took symptom development into consideration.
Mandatory warm up: stepping, sideways walk, and two-legged knee bends.Strength-training level 1: pelvic lifts (2 × 15 RM), one-legged knee bends with maximum 60° flexion (2 × 10 RM per leg), and hip abductions with elastic band (2 × 10 RM per leg).Strength-training level 2: pelvic lifts (3 × 15 RM), one-legged knee bends with maximum 60° flexion (3 × 10 RM per leg), hip abductions with elastic band (2 × 10 RM per leg), sideways slide lunges (2 × 10 RM per leg), and backward slide lunges (2 × 10 RM per leg).

### 3.3. Laser Therapy and Blinding

The laser group underwent LLLT three times per week in the first 3 weeks with an Irradia GaAs class 3B laser device in adherence to the World Association for Laser Therapy treatment recommendations for dose per treatment spot: six spots in the medial knee joint line, six spots in the lateral knee joint line, and three spots in the popliteal fossa were irradiated with super-pulsed 904 nm wavelength laser for 50 s with a mean intensity of 60 mW, resulting in 3 joules per point, that is, 45 joules per knee per session (the treatment spots are illustrated elsewhere [25]). The laser treatment was applied immediately after each supervised exercise session by the physiotherapist. The 904 nm wavelength is invisible to the naked human eye, and the low output produces no noticeable heat [27]. The participants in the placebo group were treated with a sham laser device with identical appearance, using the same procedure, but with a cut wire hidden in the machinery that resulted in no output power. This wire was cut by the manufacturer, and thus no one in the study knew which laser device was intact. The code for placebo and real LLLT were revealed after the statistical analyses were complete. These procedures ensured that the participants, therapists, assessors, and statistician were blinded to the group allocation.

### 3.4. Concomitant Interventions

The participants were allowed to receive physiotherapy for the knee during the study, but not in the intervention period (weeks 0–8). Furthermore, the participants were not allowed to receive laser therapy outside the study (weeks 0–52). The types of knee interventions made use of by participants after the intervention period were registered.

### 3.5. Outcomes

The primary outcomes were pain on movement, at night, and at rest measured with a VAS, and global pain measured with the pain subscale of the Knee injury and Osteoarthritis Outcome Score (KOOS) questionnaire. The secondary outcomes were KOOS disability, KOOS quality of life (QoL), number of participants using any analgesic and NSAIDs, global health change, knee flexion active range of motion (AROM), number of chair stands in 30 s, joint line pain pressure threshold (PPT), tibial condyle PPT, and real-time ultrasonography (RTU) findings of femur cartilage thickness, suprapatellar effusion, and meniscal neovascularization (Doppler area).

All the outcomes were assessed at baseline and 3, 8, 26, and 52 weeks later, except for global health change, which was solely evaluated at week 8, the time-point when the greatest effect of LLLT was previously observed [11]. First, the participants filled out questionnaires, then ultrasonography was performed, and finally the physical assessment was completed. The patient-reported outcome questionnaires (KOOS and VAS pain) were answered by the participants at baseline in the lab and at reassessments either in the lab or via email.

#### 3.5.1. VAS (Pain)

The VAS is a 100 mm scale that is used to score pain from 0 mm (no pain) to 100 mm (worst imaginable pain), and this tool has been found to be more reliable than the Numeric Pain Rating Scale in assessing the pain of KOA patients [28]. We chose a digital version of the VAS instead of a physical one due to convenience and since it produces the same results [29].

#### 3.5.2. KOOS (Pain, Disability, and QoL)

The KOOS questionnaire is a disease-specific tool based on Likert scales proven to be both valid and reliable, and it comprises five subscales, that is, global pain, physical function in daily living, physical function in sports and recreational activities, QoL, and other symptoms [30]. The KOOS answers were transformed to percentage scores ranging from 0–100, where a higher score is better [30].

#### 3.5.3. Global Health Change

Global health change was ranked on a 7-point scale, where a lower score is better. It was conducted by asking the participants whether they experienced no symptoms (1), a large improvement (2), some improvement (3), no change (4), some worsening (5), a large worsening (6), or worse symptoms than ever (7).

#### 3.5.4. Analgesics

The number of participants who had used any rescue analgesic (paracetamol, NSAIDs, etc.) due to knee pain 7 days prior to assessment was scored dichotomously.

#### 3.5.5. AROM

Knee flexion AROM was measured with the participants in supine position using a 2 × 30 cm goniometer, since shorter versions are not as reliable [31].

#### 3.5.6. Sit-to-Stand Test Chair Stands

The 30 s sit-to-stand test was performed to assess the physical function of the participants because this test is recommended by the Osteoarthritis Research Society International [32]. We included the final attempt when the participants were more than halfway up.

#### 3.5.7. PPT

The most tender spot on the knee joint line identified by palpation and another 1.5 cm distally from this spot (on the tibia bone) were assessed for PPT using a digital algometer (Wagner FPX 25, Greenwich, CT, USA) with a 1 cm^2^ rubber tip. The display of the algometer faced the floor during the measurements to blind the assessor and participants to the levels of force. Three PPT measurements were made, and the mean score of the final two measurements was used for analysis, since they were the most reliable [33]. The intra-rater relative reliability of the method in the joint line was found to be good in our reliability study with a convenience sample of the same participants [33].

#### 3.5.8. RTU

A RTU device (Mindray Diagnostic Ultrasound System M7, Shenzhen, China) was used to measure suprapatellar effusion and meniscal neovascularization with the knee flexed 30° and femur cartilage thickness with orthogonal probe insonation and maximum knee bend. Effusion was scored as its maximum height, meniscal neovascularization was quantified as Doppler pixel area in mm^2^, and femur cartilage thickness was measured at the medial condyle, lateral condyle, and patellofemoral groove. We corrected for cartilage thickness by including the leading interface as part of the cartilage and multiplying the results by a factor of 1.07 to account for sound traveling at different speeds in different tissues [34].

### 3.6. Statistical Analysis

The results were analyzed using the intention-to-treat approach. Both the right and left knees of the participants with bilateral and unilateral KOA were assessed, but only the osteoarthritic knees were analyzed when data allowed for it. The continuous outcome data were normally distributed according to histograms. These data were analyzed using the two-way analysis of variance (ANOVA) or ANOVA mixed model with Šidák’s correction. The short- and long-term outcome data were separated in the ANOVA (weeks 0, 3, and 8 or weeks 0, 26, and 52), since the effectiveness of LLLT has been found to vary between these time periods [11]. The significance levels of all the within-group differences were calculated using raw data in the statistical software programs. Change scores (difference between baseline and reassessment) were first calculated in data sheets and then analyzed. The ANOVA significance levels of between-group changes were calculated using change scores. The global health change data were analyzed with the Mann–Whitney U test. The between-group differences in number of participants using any analgesic and NSAIDs at individual weeks were analyzed with Fisher’s exact test, and the within-group and between-group changes in these outcomes were analyzed using the Wilcoxon signed-rank test and Mann–Whitney U test, respectively. The analyses were conducted with the software programs GraphPad Prism 9 and Software for Statistics and Data Science (STATA) 17. M.B.S. conducted the statistical analyses under supervision of J.M.B. and René B. Svensson. The power calculation was detailed in the previously published protocol [25].

## 4. Results

In total, 61 persons were assessed for eligibility for participation in the study, of which 51 met the criteria. The reason for ineligibility was that pain intensity was too low on movement. One eligible person declined to participate after the baseline assessment, but before being randomized to a group. The remaining 50 eligible persons were enrolled in the study, and 46 participants (92%) completed the study (Figure 1). In the laser group, one person dropped out after a few treatments due to illness in the family and another person did not respond to the invitation for the 52-week assessment for unknown reasons. In the placebo group, two persons did not respond to the invitation for the 26- and 52-week assessments for unknown reasons.

Pain on movement and joint line PPT were significantly worse in the placebo group than in the laser group at baseline, but no other significant baseline imbalances were detected (Table 1). Therefore, we calculated the between-group differences based on changes from baseline to the reassessment weeks (Table 2, Table 3 and Table 4). Adjusted and unadjusted within-group scores at the individual reassessment weeks and adjusted within-group significance levels for changes are reported in the Appendix A. The compliance with the intervention procedure was high in both groups. In the follow-up period, the number of weekly leg exercise training sessions of any type performed by the participants did not vary between the groups. Furthermore, there was no significant group difference in number of participants using concomitant interventions in the follow-up period (*p* = 1.00). These additional interventions were physiotherapeutic modalities, such as massage, acupuncture, and exercise therapy.

### 4.1. Within-Group Changes from Baseline

Pain on movement and global pain were statistically, significantly improved in both groups at all reassessments (Appendix A). Pain at rest was statistically, significantly improved in the placebo group at week 26 (Appendix A). Pain at night was statistically, significantly improved at weeks 3, 8, and 52 in the laser group and at weeks 3 and 8 in the placebo group (Appendix A). Patient-reported disability was statistically, significantly improved in both groups at all reassessments, except for disability in sports and recreation in the placebo group at weeks 3 and 52 (Appendix A). QoL was statistically, significantly improved in both groups at all reassessments (Appendix A). The number of participants using any analgesic was statistically, significantly reduced in the laser group at weeks 3, 8, and 52 and in the placebo group at week 8 (Appendix A). The number of participants using NSAIDs was statistically, significantly reduced in the laser group at weeks 3, 8, and 52 (Appendix A). Knee flexion AROM was statistically, significantly improved in the laser group at week 26 (Appendix A). The number of chair stands was statistically, significantly increased in both groups at all reassessments (Appendix A). Joint line PPT was statistically, significantly improved in the placebo group at weeks 8, 26, and 52 (Appendix A). No other within-group statistically significant differences were found (Appendix A).

### 4.2. Between-Group Changes from Baseline

The laser group improved statistically, significantly more regarding any analgesic and NSAID usage and chair stands at week 52 (Table 2 and Table 3). The global health change questionnaire showed that the laser group experienced a larger improvement in symptoms than the placebo group, but the difference only approached statistical significance (*p* = 0.07). The placebo group was improved statistically, significantly more regarding joint line PPT at week 8 (Table 3). No other statistically significant between-group changes were found (Table 2, Table 3 and Table 4).

## 5. Discussion

In this placebo-controlled RCT, we investigated the short- and long-term effectiveness of a high dose LLLT as a supplement to strength training. Seventeen different assessments were conducted, including patient-reported outcomes, physical tests, and RTU assessments.

### 5.1. Patient-Reported Outcomes

Patient-reported pain, disability, and QoL were generally improved in both groups throughout the study compared with baseline, but the between-group changes in these outcomes were not significant. The minimal clinically important improvement (MCII) for pain in KOA has been estimated to be 40.8% measured on a VAS [35], and in both groups this threshold was exceeded in terms of pain on movement and at night at the majority of reassessments.

Interestingly, the number of participants using any analgesic and NSAIDs specifically were reduced substantially more in the laser group than in the placebo group, and although the differences were only statistically significant at week 52, this positive trend plausibly affected the other effect estimates in a negative direction for LLLT.

At the end of LLLT (week 3), pain on movement was reduced by 51% in the placebo group, which was unexpectedly much. In our systematic review on the topic, we observed that the pain reduction in the nine placebo + exercise groups was only 20% (mean) [11]. In our RCT, the pain reduction in the LLLT + exercise group was 38%, and although this was less of an improvement than in our placebo + exercise group, it was the exact same level of pain reduction as was seen in the LLLT + exercise groups in the systematic review that demonstrated a clear superiority of LLLT over placebo [11].

### 5.2. Physical Tests

Even with the difference in usage of pain medication, the laser group was improved significantly more than the placebo group in the sit-to-stand test at week 52. Interestingly, in persons with hip osteoarthritis, the MCII in number of chair stands in 30 s has been estimated to be 2–2.6 [36], and the between-group difference at week 52 was 2.52 repetitions in favor of LLLT. This indicates that LLLT has a substantial long-term positive effect on physical performance when used in conjunction with strength training.

Joint line PPT was generally improved in the placebo group and not in the laser group, but the between-group difference in change in this outcome was only statistically significant at week 8. Furthermore, the between-group differences in tibia PPT were not significant.

Knee flexion AROM was statistically, significantly improved in the laser group, but only at week 26, and the difference in change did not differ statistically nor significantly between the groups.

### 5.3. RTU Assessments

No significant treatment effects were seen with neither suprapatellar effusion, meniscal Doppler, nor femur cartilage thickness.

### 5.4. Laser Dosing

In our systematic review on the topic, we managed to identify the lowest effective laser dose per treatment spot [11]. However, evidence regarding the optimal dose was sparse. Therefore, we made our best guess and decided to deliver a higher total dose of 904 nm of LLLT per session than in the previously published placebo-controlled RCTs on the topic [25]. In the systematic review, 904 nm laser was applied in nine trials with doses ranging from 0.2 to 27 joules per knee per session. The laser doses from 0.2 to 1.2 joules per knee were ineffective, whereas the laser doses from 2 to 27 joules per knee significantly reduced pain. Interestingly, the mean laser dose applied in the three trials with the most successful outcomes was 5.5 joules per knee per session. Therefore, the 45 joules per session with 904 nm LLLT per knee per session applied in our RCT may have been too high. Even larger laser doses have been tested out in some RCTs of high-intensity laser therapy (HILT), and they reportedly resulted in pain relief [37,38]. However, when studying the clinical effectiveness of HILT more closely, the high output power does not seem to add value convincingly. The HILT doses used induce a heat sensation in medium and highly pigmented skin [39,40] that may compromise the blinding of patients and therapists. Furthermore, in contrast to LLLT, HILT has been shown to deteriorate articular cartilage in animal models [7].

### 5.5. Strengths and Limitations

Our study featured random and concealed group allocation, and the blinding of participants, assessors, therapists, and statisticians. The drop-out rate in the study was minor (*n* = 4), even though most of the trial took place during the coronavirus pandemic. However, a substantial number of Doppler images from weeks 3, 26, and 52 were not collected due to a technical error. Furthermore, we attempted to measure the pain-free isometric knee extension strength of the participants as planned, but the limited capacity of the dynamometer used for this assessment caused a substantial ceiling effect, and thus we opted not to report these results in detail; the assessment did not show any significant group differences. The number of participants who had used analgesics were analyzed dichotomously as preplanned. Because the types of analgesics used varied, an analysis of this outcome based on continuous data, such as dose, was impossible. The usage of NSAIDs may have lowered the potential for LLLT to reduce inflammation during the entire study. At baseline, pain on movement was significantly higher and the joint line PPT was significantly lower in the placebo group than in the laser group, but no significant imbalances were seen in comparisons of the 21 other baseline variables, including pain at rest, at night, and globally and tibia PPT. However, randomization with stratification by pain intensity could have been advantageous [41]. To reduce the impact of baseline imbalances on the effect estimates, the change scores (difference between baseline and follow-up) were calculated. As pain on movement corresponding to 40 mm on the VAS was a prerequisite for participation in the study, there were no extreme outliers regarding this outcome at baseline that we could adjust for. Inflammation was measured indirectly using RTU and PPT algometry; however, these tools seemed to lack sufficient sensitivity to detect minor changes. In hindsight, we can also see that our power calculation may have been too optimistic in terms of the expected between-group difference in pain on movement of 20 mm VAS, for example. Put in perspective, this difference in change is twice as large as in RCTs of oral NSAIDs versus placebo [42]. With a powerful intervention such as exercise therapy in both groups, and a placebo laser group that improved more than in any of the previously published LLLT KOA trials [11], only a few outcomes showed a significant effect of LLLT. It is important to note that these positive differences were achieved in the long-term follow-up period.

## 6. Conclusions

Pain was reduced to a clinically relevant extent in both groups. The LLLT seemed to increase the performance in the sit-to-stand test and reduce the usage of pain medication; however, it did not significantly affect the other outcomes, including the primary outcomes. It is plausible that the LLLT dose may have been too high, since lower doses of LLLT have been applied with greater success in previous studies on the topic. The baseline imbalance in terms of more intense pain on movement and lower joint line PPT in the placebo group and the unusually large pain reduction in the placebo group may also have prevented the detection of additional LLLT treatment effects.

## Figures and Tables

**Figure 1 jcm-11-03446-f001:**
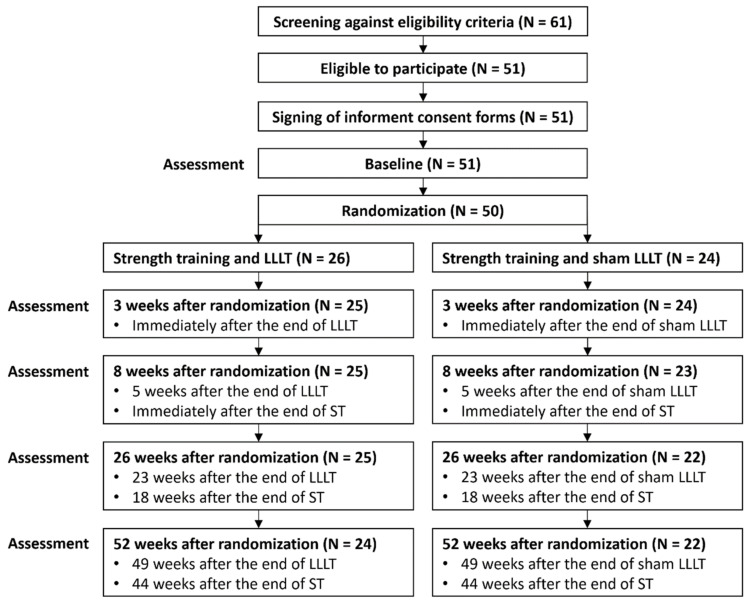
Study flowchart. Abbreviations: LLLT: low-level laser therapy; ST: strength training.

**Table 1 jcm-11-03446-t001:** Baseline characteristics of the participants in both groups.

Variables, Mean ± SD/N (%)	Laser Group	Placebo Group	*p*-Value
Age (years)	64.04 ± 8.52	61.92 ± 6.39	0.3372
Weight (kg)	83.25 ± 14.78	79.48 ± 14.30	0.3742
Height (m)	1.72 ± 0.08	1.69 ± 0.12	0.3655
BMI	28.11 ± 4.31	27.66 ± 3.58	0.6967
Gender (No.)			
Females	18 (69.23%)	19 (79.17%)	
Males	8 (30.77%)	5 (20.83%)	0.526
Duration of knee pain (months)			
Right osteoarthritic knee	92.16 ± 103.56	83.52 ± 87.63	0.7657
Left osteoarthritic knee	125.1 ± 135.83	89.18 ± 71.35	0.2899
Pain on movement (mm VAS)	52.77 ± 11.68	63.88 ± 14.87	0.0193 *
Pain at rest (mm VAS)	17.15 ± 17.17	29.63 ± 24.00	0.1325
Pain at night (mm VAS)	28.58 ± 20.61	39.29 ± 25.91	0.3233
Pain globally (KOOS)	48.61 ± 12.23	42.94 ± 14.58	0.3928
Disability in ADL (KOOS)	57.80 ± 15.18	49.25 ± 20.35	0.2923
Disability in sports/rec. (KOOS)	19.42 ± 19.82	21.88± 19.46	0.9633
Quality of life (KOOS)	25.71 ± 13.68	25.25 ± 14.26	0.9993
Users of any analgesic (N)	11 (42.31%)	9 (37.50%)	0.779
Users of NSAIDs (N)	6 (23.08%)	5 (20.83%)	1.000
Knee flexion AROM (degrees)	121.1 ± 11.08	122.0 ± 9.80	0.9863
30 s chair stands (No.)	10.23 ± 3.84	9.96 ± 3.88	0.9929
Joint line PPT (newton)	49.85 ± 20.16	32.37 ± 12.70	0.0086 **
Tibial condyle PPT (newton)	45.05 ± 21.85	34.53 ± 13.06	0.1451
Suprapatellar effusion (mm)	5.77 ± 3.595	5.01 ± 1.949	0.7624
Meniscal Doppler (mm^2^)	2.323 ± 2.28	2.713 ± 1.96	0.9520
Femur cartilage thickness (mm)	1.59 ± 0.381	1.48 ± 0.380	0.7367

Abbreviations: ADL: activities of daily living; AROM: active range of motion; BMI: body mass index; KOOS: Knee Osteoarthritis Outcome Scale; NSAIDs: nonsteroidal anti-inflammatory drugs; PPT: pain pressure threshold; rec.: recreation; VAS: Visual Analogue Scale. Significant group difference: * *p* < 0.05; ** *p* < 0.01.

**Table 2 jcm-11-03446-t002:** Patient-reported outcomes: within- and between-group changes from baseline (higher score is better).

Variables	Weeks 0–3	Weeks 0–8	Weeks 0–26	Weeks 0–52
Pain on movement (VAS)				
Laser group	20.12 (*n* = 25)	24.44 (*n* = 25)	21.76 (*n* = 25)	35.43 (*n* = 24)
Placebo group	32.29 (*n* = 24)	32.16 (*n* = 23)	35.91 (*n* = 22)	30.55 (*n* = 22)
Between-group change	−12.17 (−27.86 to 3.52)	−7.72 (−23.53 to 8.08)	−14.15 (−29.99 to 1.69)	4.88 (−11.07 to 20.85)
Pain at rest (VAS)				
Laser group	1.56 (*n* = 25)	7.88 (*n* = 25)	3.08 (*n* = 25)	8.73 (*n* = 24)
Placebo group	8.21 (*n* = 24)	9.10 (*n* = 23)	11.55 (*n* = 22)	4.64 (*n* = 22)
Between-group change	−6.65 (−21.69 to 8.40)	−1.22 (−16.37 to 13.93)	−8.47 (−24.24 to 7.31)	4.09 (−11.80 to 20.00)
Pain at night (VAS)				
Laser group	15.96 (*n* = 25)	15.60 (*n* = 25)	11.84 (*n* = 25)	22.23 (*n* = 24)
Placebo group	18.67 (*n* = 24)	21.02 (*n* = 23)	16.77 (*n* = 22)	14.18 (*n* = 22)
Between-group change	−2.71 (−19.11 to 13.70)	−5.42 (−21.89 to 11.05)	−4.93 (−25.15 to 15.29)	8.04 (−12.27 to 28.36)
Pain globally (KOOS)				
Laser group	15.00 (*n* = 25)	17.45 (*n* = 25)	17.45 (*n* = 25)	20.54 (*n* = 24)
Placebo group	14.70 (*n* = 24)	20.15 (*n* = 23)	16.67 (*n* = 22)	16.92 (*n* = 22)
Between-group change	0.30 (−9.78 to 10.38)	−2.70 (−12.83 to 7.43)	0.78 (−12.33 to 13.89)	3.62 (−9.54 to 16.77)
Disability in ADL (KOOS)				
Laser group	13.30 (*n* = 25)	15.71 (*n* = 25)	13.94 (*n* = 25)	18.92 (*n* = 24)
Placebo group	13.86 (*n* = 24)	19.41 (*n* = 23)	14.30 (*n* = 22)	12.64 (*n* = 22)
Between-group change	−0.56 (−11.04 to 9.90)	−3.70 (−14.23 to 6.83)	−0.36 (−12.93 to 12.21)	6.28 (−6.35 to 18.91)
Disability in sports/rec. (KOOS)				
Laser group	20.80 (*n* = 25)	21.60 (*n* = 25)	16.20 (*n* = 25)	20.85 (*n* = 24)
Placebo group	9.17 (*n* = 24)	15.61 (*n* = 23)	9.77 (*n* = 22)	8.86 (*n* = 22)
Between-group change	11.63 (−4.09 to 27.36)	5.99 (−9.83 to 21.81)	6.43 (−9.33 to 22.18)	11.99 (−3.84 to 27.82)
Quality of life (KOOS)				
Laser group	16.52 (*n* = 25)	21.52 (*n* = 25)	18.76 (*n* = 25)	23.36 (*n* = 24)
Placebo group	9.37 (*n* = 24)	16.01 (*n* = 23)	19.60 (*n* = 22)	16.77 (*n* = 22)
Between-group change	7.15 (−3.10 to 17.40)	5.51 (−4.81 to 15.83)	−0.84 (−12.33 to 10.64)	6.59 (−4.96 to 18.14)
Any analgesic				
Laser group	6 (24%) (*n* = 25)	6 (24%) (*n* = 25)	3 (12%) (*n* = 25)	6 (27.3%) (*n* = 22)
Placebo group	3 (12.5%) (*n* = 24)	4 (16.7%) (*n* = 24)	−1 (−4.8%) (*n* = 21)	−3 (−14.3%) (*n* = 21)
Between-group change	3 (*p* = 0.5947)	2 (*p* = 0.7802)	2 (*p* = 0.3424)	9 (*p* = 0.0127) *
NSAIDs				
Laser group	6 (25%) (*n* = 24)	5 (20.8%) (*n* = 24)	4 (16%) (*n* = 25)	5 (22.7%) (*n* = 22)
Placebo group	3 (13.0%) (*n* = 23)	3 (13.0%) (*n* = 23)	2 (9.5%) (*n* = 21)	−2 (−9.5%) (*n* = 21)
Between-group change	3 (*p* = 0.3394)	2 (*p* = 0.4514)	2 (*p* = 0.5868)	7 (*p* = 0.0234) *

Abbreviations: ADL: activities of daily living; KOOS: Knee Osteoarthritis Outcome Scale; NSAIDs: nonsteroidal anti-inflammatory drugs; rec.: recreation; VAS: Visual Analogue Scale. Between-group change from baseline is significantly different: * *p* < 0.05. Ranges are 95% confidence intervals signifying difference in change from baseline. Positive within-group change indicates improvement. Positive between-group change indicates that laser is superior to placebo.

**Table 3 jcm-11-03446-t003:** Physical assessments: within- and between-group changes from baseline (higher score is better).

Variables	Weeks 0–3	Weeks 0–8	Weeks 0–26	Weeks 0–52
Knee flexion AROM (degrees)				
Laser group	1.76 (*n* = 25)	2.72 (*n* = 25)	3.48 (*n* = 25)	2.15 (*n* = 22)
Placebo group	1.77 (*n* = 24)	2.85 (*n* = 24)	1.65 (*n* = 21)	1.52 (*n* = 21)
Between-group change	−0.01 (−3.80 to 3.78)	−0.13 (−3.93 to 3.66)	1.83 (−2.39 to 6.05)	0.63 (−3.67 to 4.92)
30 s chair stands				
Laser group	2.16 (*n* = 25)	4.08 (*n* = 25)	4.92 (*n* = 25)	5.67 (*n* = 21)
Placebo group	1.71 (*n* = 24)	3.29 (*n* = 24)	2.90 (*n* = 21)	3.15 (*n* = 21)
Between-group change	0.45 (−1.14 to 2.04)	0.79 (−0.80 to 2.38)	2.02 (−0.41 to 4.45)	2.52 (0.04 to 5.02) *
Joint line PPT (newton)				
Laser group	−4.01 (*n* = 25)	−3.66 (*n* = 25)	3.44 (*n* = 25)	2.82 (*n* = 22)
Placebo group	0.56 (*n* = 24)	9.60 (*n* = 24)	11.25 (*n* = 21)	10.56 (*n* = 21)
Between-group change	−4.57 (6.49 to −15.61)	−13.26 (−24.31 to −2.20) *	−7.81 (−20.88 to 5.26)	−7.74 (−21.11 to 5.62)
Tibial condyle PPT (newton)				
Laser group	−2.80 (*n* = 25)	−0.19 (*n* = 25)	4.30 (*n* = 25)	3.27 (*n* = 22)
Placebo group	−3.41 (*n* = 24)	5.06 (*n* = 24)	2.93 (*n* = 21)	3.70 (*n* = 21)
Between-group change	0.61 (−8.91 to 10.12)	−5.25 (−14.76 to 4.27)	1.37 (−9.86 to 12.62)	−0.43 (−11.92 to 11.05)

AROM, active range of motion; PPT, pain pressure threshold. Between-group change from baseline is significantly different: * *p* < 0.05. Ranges are 95% confidence intervals signifying difference in change from baseline. Positive within-group change indicates improvement. Positive between-group change indicates that laser is superior to placebo.

**Table 4 jcm-11-03446-t004:** RTU assessments: within- and between-group changes from baseline (higher score is better).

Variables	Weeks 0–3	Weeks 0–8	Weeks 0–26	Weeks 0–52
Suprapatellar effusion (mm)				
Laser group	−0.526 (*n* = 23)	−0.029 (*n* = 23)	0.658 (*n* = 23)	−0.119 (*n* = 21)
Placebo group	0.196 (*n* = 24)	0.331 (*n* = 23)	0.675 (*n* = 21)	0.563 (*n* = 21)
Between-group change	−0.722 (−3.106 to 1.662)	−0.360 (−2.756 to 2.036)	−0.017 (−2.204 to 2.169)	−0.682 (−2.897 to 1.531)
Meniscal Doppler (mm^2^)				
Laser group	0.145 (*n* = 17)	0.140 (*n* = 20)	0.010 (*n* = 13)	0.391 (*n* = 9)
Placebo group	0.565 (*n* = 15)	−0.783 (*n* = 18)	−0.496 (*n* = 16)	1.327 (*n* = 11)
Between-group change	−0.42 (−3.321 to 2.480)	0.923 (−1.786 to 3.632)	0.506 (−2.490 to 3.502)	−0.936 (−4.542 to 2.670)
Cartilage thickness (mm)				
Laser group	−0.099 (*n* = 23)	−0.095 (*n* = 23)	−0.093 (*n* = 22)	0.037 (*n* = 18)
Placebo group	−0.040 (*n* = 23)	0.041 (*n* = 22)	0.023 (*n* = 21)	−0.015 (*n* = 21)
Between-group change	−0.059 (−0.297 to 0.178)	−0.136 (−0.375 to 0.104)	−0.116 (−0.425 to 0.193)	0.052 (−0.269 to 0.374)

Ranges are 95% confidence intervals signifying difference in change from baseline. Positive within-group change indicates improvement. Positive between-group change indicates that laser is superior to placebo.

## Data Availability

The raw scores at reassessments are reported in the Appendix A.

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
