# Peer review of "Short- and Long-Term Effectiveness of Low-Level Laser Therapy Combined with Strength Training in Knee Osteoarthritis: A Randomized Placebo-Controlled Trial"

_jcm, 2022, doi:10.3390/jcm11123446_

Round 1
Reviewer 1 Report
The authors have thorougly revised the manuscript and implemented many of the suggested changes.
However, some points of concern remain to be resolved before publication:
Part of the text is still written in a more story-telling style than scientific style: e.g. lines 270-272 ' we registered...between the groups' This could better be stated as: the number of weekly leg exercise training sessions of any type performed by the participants did not vary between groups.
Abstract:
line 32: suggestion '... 52 weeks after initial laser or placebo treatment'
line 35: Please start the results section of the abstract with the primary outcome as it is your main finding (therefore you defined it as your primary objective).
line 39: remove was
line 45: considered 'as it reduced use of analgesics and NSAIDS'
Introduction:
The introduction could still be improved.
lines 60-97 might be condensed to better follow flow of the introduction. e.g. line 76 ' the non-recommended ... noticeable effect' could be removed because it adds no information for this topic.
results:
line 274 and 275: such as? what physiotherapeutic modalities?
line 325: please change to 'the differences in symptoms between the laser group and placebo group approached significance (p=0.07)
Conclusions:
433: please say outcomes instead of ones
436-437: made it difficult to find effects --> change to 'may have prevented the detection of additional effects'.
Author Response
We appreciate the comments by the reviewer on our manuscript. Please see our response to each comment:
Comment by reviewer 1:
“The authors have thoroughly revised the manuscript and implemented many of the suggested changes. However, some points of concern remain to be resolved before publication.”.
Our response to the comment:
We thank the reviewer for the previous and new suggestions/recommendations and have made an effort to address these.
Comment by reviewer 1:
“Part of the text is still written in a more story-telling style than scientific style: e.g. lines 270-272 ' we registered...between the groups' This could better be stated as: the number of weekly leg exercise training sessions of any type performed by the participants did not vary between groups.”.
Our response to the comment:
We have replaced the text with the following text: “In the follow-up period the number of weekly leg exercise training sessions of any type performed by the participants did not vary between the groups.”.
Comment by reviewer 1:
“Line 32: suggestion '... 52 weeks after initial laser or placebo treatment'.”.
Our response to the comment:
Done.
Comment by reviewer 1:
“Line 35: Please start the results section of the abstract with the primary outcome as it is your main finding (therefore you defined it as your primary objective).”.
Our response to the comment:
In the beginning of the results section of the abstract we now state that “There were no significant between-group differences in the primary outcomes.”.
Comment by reviewer 1:
“Line 39: remove was.”.
Our response to the comment:
Done.
Comment by reviewer 1:
“Line 45: considered 'as it reduced use of analgesics and NSAIDS'.”.
Our response to the comment:
We agree with the reviewer that it should be explained why “LLLT tended to provide a positive effect in the long-term.” We now state that “LLLT seemed to provide a positive add-on effect in the follow-up period in terms of reduced pain medication usage and increased performance in the sit-to-stand test.”.
Comment by reviewer 1:
“The introduction could still be improved.
Lines 60-97 might be condensed to better follow flow of the introduction. e.g. line 76 ' the non-recommended ... noticeable effect' could be removed because it adds no information for this topic.”.
Our response to the comment:
We have reduced the amount of text by replacing it with the following: “The recommended doses provided a substantial pain-relief beyond placebo at therapy week 4-8 and at follow-ups 2-8 weeks after completed therapy. The non-recommended (lower) doses provided no or little positive effect.“. We believe it is important to explain that a dose-response relationship was found by mentioning the lower doses as well.
Comment by reviewer 1:
“Line 274 and 275: such as? what physiotherapeutic modalities?”.
Our response to the comment:
We now state that “These additional interventions were physiotherapeutic modalities, such as massage, acupuncture, and exercise therapy.“.
Comment by reviewer 1:
“Line 325: please change to 'the differences in symptoms between the laser group and placebo group approached significance (p=0.07).”.
Our response to the comment:
We have replaced the text with the following: “The global health change questionnaire showed that the laser group experienced a larger improvement in symptoms than the placebo group, but the difference only approached statistical significance(P = 0.07).“.
Conclusions:
Comment by reviewer 1:
“433: please say outcomes instead of ones.”.
Our response to the comment:
We now state “it did not significantly affect the other outcomes, including the primary outcomes”.
Comment by reviewer 1:
“436-437: made it difficult to find effects --> change to 'may have prevented the detection of additional effects'.”.
Our response to the comment:
The text has been replaced with the following “… may also have prevented the detection of additional LLLT treatment effects.“.
Reviewer 2 Report
The only significant difference between the two groups was NSAIDs usage and chair stand times at week 52, which favored the laser group. However, there was no significant difference between groups in the usage of analgesics at weeks 3, 8, 26, and 52, in the use of NSAIDs at weeks 3, 8, and 26, and in the primary outcome of pain in all three subscores and secondary outcome of KOOS in all four subscores. I have come comments as listed below:
1. The authors claimed that “the number of participants using any analgesic and NSAIDs specifically were reduced substantially more in the laser group than in the placebo group, and although the differences were only statistically significant at week 52, this positive trend plausibly affected the other effect estimates in a negative direction for LLLT”. However, according to Table S1, there was only a statistically significant difference between the two groups noted at week 52 in the usage of NSAIDs, and there was no difference in analgesics between the two groups. Furthermore, only 6 (23%) and 5 (20%) participants initially received NSAIDs in the laser and placebo groups. And at the same time, up to 3 and 2 participants had taken analgesics and NSAIDs simultaneously in the laser and placebo groups, respectively (the author’s response in the letter). Therefore, the number of medication usage was too small to conclude. No further analysis of the number of changes in both medication usage, total used days of medication, complications, compliance of drugs, etc., would also affect the interpretation of the results.
2. How to explain why there was no between-groups difference at week 0-3, week 0-8, and week 0-26 for the chair stands times, but the improvement was noted at week 0-52, especially since there was no difference in the improvement in the primary outcome of pain in all three subscores and secondary outcome of KOOS in all four subscores, including the pain subscore, between the two groups in the whole course?
3. The primary outcome for pain during movement, rest, and night showed no significant difference in improvement between the two groups. The secondary outcome of KOOS pain, ADL, sports and recreation, and quality of life showed no significant improvement between the two groups. The two groups' only considerable differences were NSAIDs usage and chair stand times at week 52, which favored the laser group. Therefore, the authors could not conclude that “LLLT tended to provide a positive add-on effect in the long-term.”
4. Please explain no statistical improvement between groups in analgesic usage at weeks 3,8,26, and 52 and NSAIDs use at weeks 3,8 and 26. Still, a significant difference between groups was noted in NSAIDs usage at week 52.
5. Please provide the data of KOOS-other symptoms of both groups during the whole follow-up period and their between-group differences.
6. The data errors prescribed in the previous comment persisted in spite that I had pointed out where the mistakes were present. Detailed data check-up was the responsibility of the authors before re-submitting the text. According to Table S1, the data in Table 2 (any analgesic and NSAIDs) should be the number I had high light, and there was a lack of the numbers of changes between-group in NSAIDs.
I had attached two tables for reminding authors where the mistakes were presented in the text and tables by the online system.

Author Response
We appreciate the comments by the reviewer on our manuscript. Please see our response to each comment:
Comment by reviewer 2:
“The only significant difference between the two groups was NSAIDs usage and chair stand times at week 52, which favored the laser group. However, there was no significant difference between groups in the usage of analgesics at weeks 3, 8, 26, and 52, in the use of NSAIDs at weeks 3, 8, and 26, and in the primary outcome of pain in all three subscores and secondary outcome of KOOS in all four subscores. I have come comments as listed below:”.
Our response to the comment:
We have stated the number of participants who used any analgesics and NSAIDs for each reassessment week in the supplementary file (Table S1) and the changes in these numbers (difference between baseline and reassessment weeks) in the main manuscript (Table 2). The reviewer has pointed out that some of these numbers were entered incorrectly in Table 2. We have now checked the tables for correctness. We agree with the reviewer and have corrected the numbers. The analysis of these outcomes was done using the correct data from Table S1 and, therefore, the P-values remain the same.
Comment by reviewer 2:
“1. The authors claimed that “the number of participants using any analgesic and NSAIDs specifically were reduced substantially more in the laser group than in the placebo group, and although the differences were only statistically significant at week 52, this positive trend plausibly affected the other effect estimates in a negative direction for LLLT”. However, according to Table S1, there was only a statistically significant difference between the two groups noted at week 52 in the usage of NSAIDs, and there was no difference in analgesics between the two groups. Furthermore, only 6 (23%) and 5 (20%) participants initially received NSAIDs in the laser and placebo groups. And at the same time, up to 3 and 2 participants had taken analgesics and NSAIDs simultaneously in the laser and placebo groups, respectively (the author’s response in the letter). Therefore, the number of medication usage was too small to conclude. No further analysis of the number of changes in both medication usage, total used days of medication, complications, compliance of drugs, etc., would also affect the interpretation of the results.”.
Our response to the comment:
It is correct that only the number of participants using NSAIDs at week 52 differed significantly between the groups (Table S1). However, due to baseline imbalances we opted to focus on the changes from baseline to the reassessment weeks, and there were significant between-group differences in change from baseline to week 52 in both any analgesic and NSAID usage (Table 2).
In the previous peer-review report the reviewer asked us to provide the number of participants using non-antiinflammatory and NSAIDs simultaneously. We agree with the reviewer that the numbers are neglectable and cannot form the basis of a conclusion.
We agree with the reviewer that a more detailed analysis of pain medication usage could have strengthened a conclusion. In the ‘Strengths and Limitations’ section we have added that “The number of participants who had used analgesics was analyzed dichotomously as pre-planned [1]. Because the types of analgesics used varied, an analysis of this outcome based on continuous data, such as dose, was impossible.”.
Comment by reviewer 2:
“2. How to explain why there was no between-groups difference at week 0-3, week 0-8, and week 0-26 for the chair stands times, but the improvement was noted at week 0-52, especially since there was no difference in the improvement in the primary outcome of pain in all three subscores and secondary outcome of KOOS in all four subscores, including the pain subscore, between the two groups in the whole course?”.
Our response to the comment:
There was a trend that the number of chair-stands had increased more in the laser group than in the placebo group at all reassessments, although it was only significant at week 52. The same was seen with both any analgesics and NSAIDs.
Comment by reviewer 2:
“3. The primary outcome for pain during movement, rest, and night showed no significant difference in improvement between the two groups. The secondary outcome of KOOS pain, ADL, sports and recreation, and quality of life showed no significant improvement between the two groups. The two groups' only considerable differences were NSAIDs usage and chair stand times at week 52, which favored the laser group. Therefore, the authors could not conclude that “LLLT tended to provide a positive add-on effect in the long-term.”.
Our response to the comment:
As mentioned, there was a trend that the laser group improved more than the placebo group throughout the study in terms number of chair-stands, any analgesics and NSAIDs. Although the analyses of these outcomes only reached statistical significance at week 52, the trend does allow for a positive conclusion, in our opinion. We agree with the other reviewer (reviewer 1) that a positive conclusion can be supported.
Comment by reviewer 2:
“4. Please explain no statistical improvement between groups in analgesic usage at weeks 3,8,26, and 52 and NSAIDs use at weeks 3,8 and 26. Still, a significant difference between groups was noted in NSAIDs usage at week 52.”.
Our response to the comment:
There was a trend that the laser group improved more than the placebo group throughout the study in terms number of chair-stands, any analgesics and NSAIDs. These three outcomes showed a significant group difference at week 52 (Table 2-3).
Comment by reviewer 2:
“5. Please provide the data of KOOS-other symptoms of both groups during the whole follow-up period and their between-group differences.”.
Our response to the comment:
We prefer not to analyze KOOS other symptoms because it was not pre-planned, the group codes for laser and placebo have been revealed, the manuscript includes unusually many outcomes and time-points of assessments, and the outcome is arguably not as interesting as the other patient-reported outcomes.
Comment by reviewer 2:
“6. The data errors prescribed in the previous comment persisted in spite that I had pointed out where the mistakes were present. Detailed data check-up was the responsibility of the authors before re-submitting the text. According to Table S1, the data in Table 2 (any analgesic and NSAIDs) should be the number I had high light, and there was a lack of the numbers of changes between-group in NSAIDs.”.
I had attached two tables for reminding authors where the mistakes were presented in the text and tables by the online system.”.
Our response to the comment:
We thank the reviewer for making us aware of the errors in Table 2. These errors have been corrected. We apologize for overlooking the attached file with the descriptions of the errors in the previous peer-review round.
This manuscript is a resubmission of an earlier submission. The following is a list of the peer review reports and author responses from that submission.
Round 1
Reviewer 1 Report
This study evaluated both short and long-term effects of low-level laser therapy associated with a strength training exercise in knee OA. I have come comments as listed below:
- Although the authors had pointed out that the power calculation was too optimistic in the present study, they did not calculate the sample size in the present study.
- The trial registration, including the published protocol, was retrospectively registered after the completion of the trial.
- The numbers of participants with analgesics and NSAID were reduced significantly in the laser treatment group than the placebo group at week 52. Please give us the information related to (1) dosage of analgesics and NSAID, and (2) duration of analgesics and NSAID, how long they took the medication at the follow-up time.
- Please provide numbers of participants who took analgesics and NSAID simultaneously.
- Please give us the other treatment provided in the study periods in the two groups.
- Regarding the chair stands, how to explain why there was no inter-group difference during the whole course of treatment and follow-up at 26 weeks periods, but significant improvement of stairs stand at six months in the laser group compared to the placebo group at 52 weeks?
- How to explain that no significant improvement of pain in the laser group compared to the placebo group during the treatment and whole follow-up periods up to 52 weeks, but significant improvement of chair stand performance at 52 weeks in the laser group?
- There were errors of number listed in Table 2, including the number of analgesic at weeks 0-26 in the placebo group and NSAID in the placebo group in weeks 0-3, 0-8, 0-26, and 0-52.
Reviewer 2 Report
It is important to perform randomized controlled trials and to publish these, even if negative results are obtained. The authors have pre-published the trial protocol, which is a good way to carry out open science. This study has some important flaws in the study design, most importantly that it is not sufficiently powered to demonstrate small differences. The authors acknowledge this in their discussion.
Furthermore, the language and style of the manuscript are often unscientific and could be much improved. some more specific comments are provided below.
Line 29: self reported ? Do you mean questionnaires/ PROMS? Or self reported physical assessments?
Line 32: first outcomes that are described are use of NSAIDS and analgesics while these are not described in methods. The methods should state clearly what primary outcome is and the result should mention this first. It is important to note that there was no significant difference in the primary outcome as defined in the published protocol. This should be the main finding of the paper.
Line 37: describe noteworthy worse condition more specifically
Introduction:
It is a bit extensive and the story is not easy to follow. There are too many abbreviations used. The introduction also reads rather unscientific.
E.g. We conducted (line 79)… hence we decided to investigate (line 84)
Might be better stated as: long-term evaluations of … are currently lacking in literature, therefore this study aims to …
Results:
It should be noted that this study was powered to find a difference in pain during movement of 20 mm VAS, which is a very big difference for this kind of treatment. There might be a difference favoring LLLT but this study is underpowered to find it. My major concern is that the authors focus on their secondary outcomes such as the medicine usage and physical exams.
The fact that there are significant and substantial differences in pain and joint line tenderness at baseline, even though blinded randomization was carried out, is worrisome and can make it hard to draw firm conclusions.
Line 300, p 0.06 is not significant.
Figure 1 flowchart: how many patients were screened for eligibility, how many were eligible and declined inclusion. These nrs are important to report.
Discussion:
How does this study improve understanding of pathophysiology and tissue morphology? This is a very broad remark and the results do not describe any such findings. Describe more specifically what you did and did not find.
The remarks as stated in lines 313-320 are very important but might be better positioned a bit further on in the discussion (after a summary of findings). The discussion repeats a lot of findings rather than putting them in perspective in light of the existing literature or discuss the generalizability etc. Therefore the discussion adds little depth to the manuscript.
Even though it is described that a statistician was involved in the study, the fact that statistics are performed on secondary outcomes while the primary outcome wasn’t significant seems unconventional. Moreover, the use of other statistical methods like mixed models could allow correction for baseline pain and solve the problems with differences at baseline and handle missing data.
Overall, I think the manuscript would benefit greatly if the statistics would be re-evaluated and the manuscript re-written with a focus on primary outcomes at first, descriptions of findings in secondary outcomes and a more in depth discussion about the findings.
Reviewer 3 Report
Stausholm et al. submitted the paper entitled “Short- and long-term effectiveness of low-level laser therapy combined with strength training in knee osteoarthritis: A randomized placebo-controlled trial”. Overall, the results show a new adjuvant therapy to the nonoperative treatment of knee osteoarthritis. Although the important work was presented by the authors, the article needs to be revised to be considered for publication in the Journal of Clinical Medicine. Please find the specific comments below.
Line 24: The sentence “Investigate the short- and long-term placebo-controlled effectiveness of LLLT combined” sounds as if the placebo has an effect of LLLT. Please revise English.
Line 26: “Fifty participants were enrolled. LLLT or placebo LLLT was performed triweekly for 3 weeks.” Please revise English so that the reader is clear on which group received what.
Line 27: Total number of participants are mentioned but you do not mention how many have been allocated to the groups.
Line 30: “0,3,8,26, and 52 weeks post-randomization” Reconsider wording as there could be a time lapse between randomization and initiation of therapy or placebo.
Line 32-36: “Usage of any… but only significantly at week 8.” Improving usage is open to interpretation, please be more precise with your English. Please reformulate the statement.
Line 39: “Clinically relevant”. This is open to interpretation, a clear distinction should be made, reconsider wording.
Line 47-48: “Tissue damage, dysfunctional metabolism, and the immune system [2]”. The types of OA you have listed are subsets or phenotypes of OA, while the term “immune system” can refer to rheumatoid arthritis. Furthermore, the reference listed only makes reference to the metabolic phenotype. Please consider substantial rewording and include the work “Classification of patients with knee osteoarthritis in clinical phenotypes: Data from the osteoarthritis initiative” by Dell’Isola et al.
Line 56: “demonstrated that laser may have a protective effect” - Laser therapy?
Line 65-66: “from therapy week 4-8 through follow-ups 6-8 weeks after completed LLLT,” Please revise English.
Line 98: Equal sex distribution is not addressed, nor is the number of total participants. “in the age of > or = 50 years“. Please revise poor English.
Line 101: The American college of rheumatology clinical criteria for knee osteoarthritis is heavily based on clinical presentation and patient self-reporting, please include a paragraph explaining whether the pre-trial screening had any radiological assessment (i.e Kellgren-Lawrence classification) to potentially exclude meniscal and/or ligament damage which might cause similar symptoms to knee osteoarthritis. Also, a clear explanation of whether or not bilateral KOA was an exclusion criterion.
Line 102: Alloplastic incorrect terminology
Line 116-119: “triweekly for the first eight weeks. The exercises were performed under supervision of a physiotherapist in the university outpatient clinic three times per week in the first three weeks and once per week in the subsequent five weeks of the study (15 supervised and 9 unsupervised exercise sessions).” supervision: 3x3weeks=9 sessions 1x5=5 9+5=14 supervised Unsupervised: 2x5weeks=10 Math doesn't add up.
Line 156: “Other types of knee interventions” Including NSAID usage?
Line 167: It is not clearly stated whether participants kept up with physical therapy during the follow-up period. Also, when was LLLT therapy stopped?
Line 184: “0-100 percentages” Please revise the English.
Line 207: “Three measurements were made and the mean of the two last ones were used for analysis” Sentence unclear please revise English.
Line 216: This reference is oddly placed and refers to vascularization of Achilles tendon rupture. I see no clinical relevance for it for your article.
Line 308: “This study has improved our understanding of what occurs with the local pathophysiology, tissue morphology” Wishful thinking. Your results are in line with that LLLT is a therapeutic tool. But it does not offer any insight into the pathophysiology of osteoarthritis nor its tissue morphology. Please revise
Line 389: “To encounter baseline imbalances, the changes from baseline were calculated” Train of thought unclear, please revise.
Line 398-400: “The unusually large pain reduction in the placebo group may also have made it difficult to find additional effects of LLLT” Do you have a theory as to why?
In summation, before considering the paper for publication, minor revision is needed as the English lacked clarity in many aspects. Please find a native English speaker to review the paper. Furthermore, the methods need to be better explained to not muddle the clarity of the results. Lastly, please include radiological screening in your methods.